# Assessment of Post-Vaccination Antibody Response Eight Months after the Administration of BNT1622b2 Vaccine to Healthcare Workers with Particular Emphasis on the Impact of Previous COVID-19 Infection

**DOI:** 10.3390/vaccines9121508

**Published:** 2021-12-20

**Authors:** Blanka Wolszczak-Biedrzycka, Anna Bieńkowska, Justyna Dorf

**Affiliations:** 1Department of Psychology and Sociology of Health and Public Health, University of Warmia and Mazury in Olsztyn, Warszawska 30, 10-082 Olsztyn, Poland; anna.bienkowska@uwm.edu.pl; 2The Oncology Center of the Region of Warmia and Mazury in Olsztyn, Hospital of the Ministry of the Interior and Administration, Wojska Polskiego 37, 10-228 Olsztyn, Poland; 3Department of Clinical Laboratory Diagnostics, Medical University of Białystok, Waszyngtona 15A, 15-269 Bialystok, Poland; justyna.dorf@umb.edu.pl

**Keywords:** COVID-19, anti-SARS-CoV-2 S antibodies, BNT162b2, humoral immune response

## Abstract

At the end of 2020, COVID-19 vaccination programs were initiated in many countries, including Poland. The first vaccine approved in Poland was the BNT162b2 mRNA preparation (Pfizer/BioNTech), and the first vaccinated group were healthcare workers. The aim of the present study was to evaluate post-vaccine antibody titers 8 months after the second vaccine dose had been administered to a group of employees of the Hospital of the Ministry of the Interior and Administration in Olsztyn (Poland). The employees were divided into two groups: persons who had COVID-19 in the fourth quarter of 2020 and were vaccinated in January–February 2021, and persons without a history of COVID-19 who were vaccinated during the same period. The analyzed material was venous blood serum collected from 100 hospital employees on 23–28 September 2021. The level of anti-SARS-CoV-2 S antibodies was measured with a Roche Cobas e411 analyzer using the electrochemiluminescence (ECLIA) method. The study demonstrated that persons with a history of SARS-CoV-2 infection had significantly higher antibody levels (taking into account gender, age, type of work performed, and severity of post-vaccination symptoms) than employees without a history of COVID-19. The study also revealed that the type of work, age, gender, and the course of SARS-CoV-2 infection can influence the humoral immune response. The presented results may prove helpful in the context of administering additional vaccine doses.

## 1. Introduction

COVID-19, the disease caused by the severe acute respiratory syndrome coronavirus (SARS-CoV-2), was classified as a pandemic by the World Health Organization on 11 March 2020 [1]. The elderly, persons with multiple comorbidities and healthcare workers, mainly medical personal, are at higher risk of infection and a more severe course of the disease [2]. The initially undertaken preventive measures, including social distancing, the use of personal protection equipment and frequent disinfection of hands, decrease the public health, social and economic impact of the pandemic only partially [3]. Moreover, it remains unknown whether having undergone infection with SARS-CoV-2 protects against future illness and if so, for how long [4]. It has been shown that in infected persons, antibodies appear already three days after the occurrence of symptoms and achieve a maximum level at 7–14 days. IgM antibody levels peak between 14 to 35 days after infection and decrease within the next 21–35 days. IgG antibodies, on the other hand, attain the highest levels approximately 21 to 49 days after infection and persist in the blood for up to 4 months [5,6]. Research into post-vaccine antibody levels that confer effective protection against the disease and produce a lasting immune response is essential for understanding the body’s defense mechanisms and may help in developing effective treatment against COVID-19 [6,7,8]. Since the beginning of the pandemic, scientists from all countries have been searching for a vaccine that would significantly reduce the number of new cases as well as the severity of the disease and the risk of hospitalization.

In Poland, BNT162b2 mRNA (Pfizer/BioNTech) was one of the first COVID-19 vaccines approved for use in persons older than 16 [9]. BNT162b2 is an mRNA vaccine which comprises mRNA encoding full-length SARS-CoV-2 virus spike glycoprotein (S protein) in the form of lipid nanoparticles (LNP). After vaccination, mRNA is translated to S protein which is then expressed on the surface of host cells. The foreign protein is recognized by the immune system, which leads to the production of neutralizing antibodies and induces a cellular response [10]. Research conducted to date has confirmed the vaccine’s high efficacy and safety [9,11]. Already 7 days post-vaccination, immunity was estimated at 68% and increased to around 93% 14 days post-vaccination. The highest, 95% effectiveness was attained 7 days after the second vaccine dose [12,13].

Post-vaccine antibody titers can be a significant predictive factor in forecasting a vaccine’s long-term efficacy, and it can be helpful in optimizing the vaccination strategy. Research conducted in the USA has shown that in people who had been infected with COVID-19, a single dose of the B162b2 vaccine conferred similar immunity to that noted in persons who had not been ill and received two vaccine doses [14,15]. However, the body’s immune response to vaccination has not been fully elucidated to date, and the most effective vaccination strategy has not been identified. There is also a general scarcity of information about the short-term and long-term effects of vaccination and antibody persistence [16]. In Poland, a third vaccine dose is recommended minimum 6 months after receiving the second dose.

The aim of this study was to evaluate the levels of anti-SARS-CoV-2-S antibodies in hospital employees in Olsztyn (Poland) 8 months after the administration of two doses of the B162b2 vaccine. In addition, antibody levels were compared in subjects who were divided into groups based on age, gender, type of work performed, history of SARS-CoV-2 infection, as well as the severity of disease symptoms and post-vaccination symptoms.

## 2. Materials and Methods

The study was approved by the Bioethics Committee of the Warmian-Masurien Medical Chamber in Olsztyn, Poland (permission number 36/2021/VIII). After a detailed explanation of the purpose of our research and the possible risk, all the qualified patients agreed in writing to participate in the experiment. The research was conducted in accordance with the World Medical Association Declaration of Helsinki for ethical principles for medical research involving human subjects.

### 2.1. Study Group

The study group consisted of 100 employees of the Hospital of the Ministry of the Interior and Administration in Olsztyn, aged 25–67, both men and women, who received the second dose of the B162b2 vaccine between 25 January and 17 February 2021. All volunteers signed informed consent forms to have their blood collected and participate in the study.

The participants were divided into two groups: persons who were vaccinated and had been previously infected with SARS-CoV-2 (*n* = 50)—confirmed by a positive PCR test performed with a Gene Expert System (Cepheid) confirming the expression of two genes—E and N2, and persons who were vaccinated and had not been previously infected with SARS-CoV-2 (*n* = 50). The minimal time in the study group between positive PCR test and second dose of vaccination was 48 days whereas maximum was 146 days, therefore patients have been divided into two groups: before and after 100 days between both dates (Table 1).

All participants gave their personal data and described the severity of post-vaccination symptoms in a questionnaire. Persons with a history of infection also gave the date of a positive PCR test and described the severity of COVID-19 symptoms. The two groups were divided into subgroups by gender (male, female), age (≤50, >50), type of work performed (medical, non-medical) and symptoms after receiving the second dose (mild to moderate and severe). Convalescents were additionally divided into 2 groups based on the severity of symptoms (mild to moderate and severe).

Blood samples for analyses of antibody levels were collected between 23 and 28 September 2021, i.e., approximately 8 months after the second vaccine dose. To obtain serum samples, blood was collected into Vacutainer test-tubes with a red stopper. Blood samples were centrifuged for 10 min at 4000× *g* at room temperature. The serum was separated and frozen at −80 °C until analysis.

### 2.2. Determination of Antibody Levels

Anti-SARS-CoV-2-S antibody levels were measured with a Cobas e411 analyzer (Roche Diagnostics, Basel, Switzerland) using the electrochemiluminescence (ECLIA) method. The analyzer supports the quantitative determination of antibodies (including IgG) to the SARS-CoV-2 spike (S) protein receptor binding domain (RBD) in human serum. The assay uses a recombinant protein representing the RBD of the S antigen in a double-antigen sandwich assay format, which favors detection of high affinity antibodies against SARS-CoV-2. The test is intended as an aid to assess the adaptive humoral immune response to the SARS-CoV-2 S protein. The measured threshold is ≥0.4 U/mL, and values ≥0.8 U/mL are considered positive. Samples with titers of >250 U/mL were diluted 10x at a time until the titer became ≤250 U/mL, according to the manufacturer’s protocol [17].

### 2.3. Statistical Analysis

Statistical analysis was performed using the GraphPad Prism (GraphPad Software, La Jolla, CA, USA). The Shapiro-Wilk test was used to examine the distribution of results. For a normal distribution, the Student’s *t*-test was used. In the case of the lack of normal distribution, the Mann-Whitney U test was used. For multiple comparisons, the ANOVA test was used with Tukey’s post hoc test, or the ANOVA Kruskal-Wallis test followed by the Dunn test. The data were presented as median (minimum–maximum).

## 3. Results

### 3.1. Characteristics of the Study Group

The study involved 100 vaccinated hospital employees. The participants were divided into two groups: persons with a history of COVID-19 (50 persons) and without a history of COVID-19 (50 persons). The majority of vaccinated convalescents were women (82%), medical personnel (86%) and persons older than 50 (36%). The majority of employees without a history of SARS-CoV-2 infection were also women (92%), medical staff (58%) and persons older than 50 (52%). The studied groups are described in detail in Table 1.

### 3.2. Comparison of Total Anti-SARS-CoV-2 Antibodies Level in Group of Workers without History of COVID-19

We observed statistically significant increase in total anti-SARS-CoV-2S antibodies level in the group of women compared to men without history of COVID-19 (*p* = 0.0239) (Figure 1A). The total anti-SARS-CoV-2S antibodies level was also significantly higher in the group of medical in comparison with non-medical workers without history of COVID-19 (*p* = 0.0088) (Figure 1D).

### 3.3. Comparison of Total Anti-SARS-CoV-2 Antibodies Level in Group of Workers with History of COVID-19

We demonstrated considerably higher level of total anti-SARS-CoV-2 antibodies in group of women than in group of men with history of COVID-19 (*p* = 0.0230) (Figure 2A). Simultaneously, the level of total anti-SARS-CoV-2 antibodies was increased in patients before 50 in comparison to those after 50 years old with history of COVID-19 (*p* = 0.0335) (Figure 2B). The total anti-SARS-CoV-2 antibodies level after vaccination was higher in patients with severe symptoms during COVID-19 than in patients with mild to moderate symptoms (*p* = 0.0172) (Figure 2D) as well as in the group of medical workers compared to non-medical with history of COVID-19 (*p* = 0.0088) (Figure 2F).

### 3.4. Comparison of Total Anti-SARS-CoV-2 Antibodies Level between the Groups with and without History of COVID-19

Total anti-SARS-CoV-2 antibodies level after vaccination was significantly higher both in the group of women (*p* < 0.0001) and men (*p* = 0.0010) after COVID-19 compared to the patients without history of COVID-19 (Figure 3A). We also demonstrated considerably higher level of total anti-SARS-CoV-2 antibodies both in group of in patients before 50 (*p* < 0.0001) and after 50 years old (*p* < 0.0001) with history of COVID-19 in comparison to those without history of COVID-19 (Figure 3B). Total anti-SARS-CoV-2 antibodies level was significantly increased in group of patients with history of COVID-19 with mild to moderate 50 (*p* < 0.0001 and severe symptoms (*p* = 0.0101) after vaccination than in patients with the same symptoms but without history of COVID-19 (Figure 3C). Medical and non-medical workers after COVID-19 had a considerably higher level of total anti-SARS-CoV-2 antibodies in comparison with workers without history of COVID-19 (Figure 3D).

## 4. Discussion

Considerable research on vaccine-elicited anti-SARS-CoV-2 antibody profiles has been done in recent months. Antibody levels are assessed in view of the time that elapsed from the administration of the first and second dose of the vaccine, as well as demographic factors and history of COVID-19 infection [18].

By the end of the third quarter of 2021, vaccination had been provided to more than 3 milliard people worldwide, i.e., less than 38% of the global population. In Poland, only 52% of the population (20 million people) had received two vaccine doses [19]. Starting in November 2021, all vaccinated persons will be eligible for a third dose (second dose in the case of the Johnson & Johnson vaccine), and the high-risk population, including healthcare workers, became eligible in October 2021. The Pfizer/BioNTech COVID-19 vaccine demonstrates 95% efficacy against the severe form of the disease in persons without a history of COVID-19. However, the persistence of the immune response after vaccination remains insufficiently investigated [20]. According to the information published at https://www.gov.pl/web/szczepimysie (accessed on 4 November 2021), vaccine-induced immunity persists for around one year. However, the current recommendation is to receive the follow-up shot at least 6 months after the completion of the first vaccination cycle (two doses of the Pfizer/BioNTech, Astra Zeneca, and Moderna vaccines or one dose of the Johnson & Johnson vaccine) [19].

Our research demonstrated that 8 months after the administration of two doses of the B162b2 vaccine, anti-SARS-CoV-2 S antibodies were still detectable, and total antibody titers were considerably higher than the limit of detection of the Elecsys anti-SARS-CoV-2-S test (cut-off value of 0.4 U/mL). The average antibody titer was 2162 U/mL in the group of employees with a history of SARS-CoV-2-S infection (confirmed by a positive PCR test), and 522 U/mL in the group without a history of COVID-19. A similar study was conducted by Campo et al. [21] on a group of 274 healthcare workers. In their study, the average antibody titer 6 months after the administration of two doses of the Pfizer/BioNTech vaccine was 130.81 AU/mL in persons without a history of COVID-19, and it was higher at 211.14 AU/mL in vaccinated convalescents. Doria-Rossa et al. [22] also demonstrated the presence of antibodies 6 months post-vaccination. In other studies evaluating antibody levels in healthcare workers, antibodies were detected 3 months after the completion of the full vaccination cycle, and antibody titers were significantly above the cut-off value [23,24].

Antibody titers were 4.23 times higher in vaccinated convalescents than in vaccinated persons without a history of COVID-19. Significant differences were also observed between persons with and without a history of COVID-19 infection who were divided into groups based on gender, age, type of work performed, and severity of post-vaccination symptoms. Similar observations were made by researchers at the Nicolaus Copernicus University in Toruń [20] as well as Angyal et al. [25] who also studied healthcare workers and demonstrated that antibody titers were 6.8 times higher in convalescents who had received the Pfizer/BioNTech vaccine than in vaccine recipients without a history of COVID-19 infection [20,25]. Vaccinated convalescents also developed a much stronger humoral immune response in the research conducted by Gobbi et al. [26], Mazzoni et al. [27], Morales-Nunez et al. [28], Ebinger et al. [14], Campo et al. [21] and Favresse et al. [24]. These observations significantly contribute to the discussion on the optimal timing of the booster shot in this group.

According to many studies, already the first vaccine dose induces a very strong humoral immune response in convalescents [29,30]. Krammer et al., demonstrated that in persons with a history of COVID-19, IgG antibody levels induced by the first Pfizer vaccine were 10 to 45 times higher in convalescents than in persons who had not been infected with the SARS-CoV-2 virus. Antibody titers were also higher in convalescents who had received the first vaccine than in persons with no history of COVID-19 who had received two doses. In addition, the administration of the second vaccine to convalescents did not cause a significant increase in antibody levels [30].

In the present study, in the group of participants without a history of COVID-19 who had received two doses of the Pfizer/BioNTech vaccine, antibody titers were considerably higher in women than in men and in medical staff than in non-medical employees. Similar conclusions were formulated by Salvagno et al. [31] and Dörschug et al. [32] who reported that women produced 1.15–1.2 times more antibodies than men on average.

However, in the group of vaccinated convalescents, antibody titers were higher in men, and the differences between genders were statistically significant. This observation suggests that the relationship between gender and humoral immune response to vaccination is not significant. In the group of employees with a history of SARS-CoV-2 infection, a significant increase in antibody levels was affected by age, severity of disease symptoms, and type of work performed. Persons older than 50, severe COVID-19 patients, and medical employees developed a stronger humoral immune response to the B162b2 vaccine. In the group of vaccinated participants without a history of COVID-19, antibody levels were also higher in persons older than 50, but the noted difference was not statistically significant. Müller et al. [33] evaluated age-associated differences in the humoral immune response (<60 years and >80 years) of persons who had received one and two doses of the vaccine. They found that antibody titers were lower in the 80+ group than in subjects younger than 60. In addition, after the second vaccine dose, 31.3% of the participants in the 80+ group had not developed neutralizing antibodies, whereas in the younger group, antibodies were absent in only 2.2% of the subjects. Tretyn et al. [20] and Angyal et al. [25] found no correlations between the participants’ gender and age and post-vaccination antibody titers. Meanwhile, other researchers [31,34,35,36,37] observed a weaker vaccine response in the 60+ group than in persons younger than 60.

The persistence of post-vaccination antibodies and the duration of immunity after the administration of the Pfizer vaccine have not been accurately determined to date [12]. It should also be noted that the presence of antibodies is not always associated with full and long-term immunity against the SARS-CoV-2 infection, which is particularly important in groups with increased exposure to the virus [21,31]. The current study confirmed that the risk of infection is particularly high among medical staff. Medical personnel accounted for 86% of convalescents and for 56% of the participants without a history of COVID-19 infection. Moreover, in both groups, antibody titers were significantly higher in medical personnel than in non-medical staff.

The study also demonstrated that antibody levels in convalescents are dependent on the severity of disease symptoms (*p* < 0.05). The presence of a relationship between the severity of clinical post-vaccination symptoms and antibody levels was also reported in a number of studies [37,38,39,40]. Our study confirms these observations. In both groups, antibody levels were higher in persons with severe post-vaccination symptoms than in persons with mild or no symptoms, but these differences were not statistically significant.

Zhong et al. [41] and Parry et al. [42] demonstrated significantly higher antibody levels in patients with extended vaccine dosing intervals and concluded that a longer interval between infection and first vaccine dose may enhance and extend the antibody response. In our work we didn’t find significant differences in antibody levels in groups of patients in which time between infection and second dose was lower and higher than 100 days. The lack of statistical differences may result from heterogenous number of patients in both groups.

Research into the persistence of post-vaccination antibodies contributes to the optimization of the vaccination system worldwide. In other studies, antibodies were present 8 months after the administration of the second BioNTech vaccine, which is consistent with our findings. These results are encouraging, and they confirm that vaccines produce a satisfactory humoral immune response. Our findings contribute to the discussion on the best timing for a booster dose, especially in persons with a history of COVID-19. In our next study, the levels of anti-SARS-CoV-2 antibodies will be determined in the same population 11 months after the administration of the second BNT162b2 vaccine, i.e., 3 months after the period described in the present article.

## 5. Conclusions

In the examined population of healthcare employees who had received 2 doses of the Pfizer/BioNTech vaccine in the same period, anti-SARS-CoV-2 S antibody titers at 8 months post-vaccination were significantly higher in convalescents than in persons without a history of COVID-19 infection. In view of published reports on antibody titers in convalescents who had received one dose of the BNT162b2 vaccine, the reported minor increase in antibody levels after the second dose, and the long persistence of post-vaccine antibodies (8 months in this study), it can be concluded that antibody levels should be checked before the administration of the booster dose to determine the body’s humoral immune response. In population groups at very high risk of exposure to the virus, antibody levels should be monitored as an auxiliary parameter for optimizing the vaccination strategy.

## Figures and Tables

**Figure 1 vaccines-09-01508-f001:**
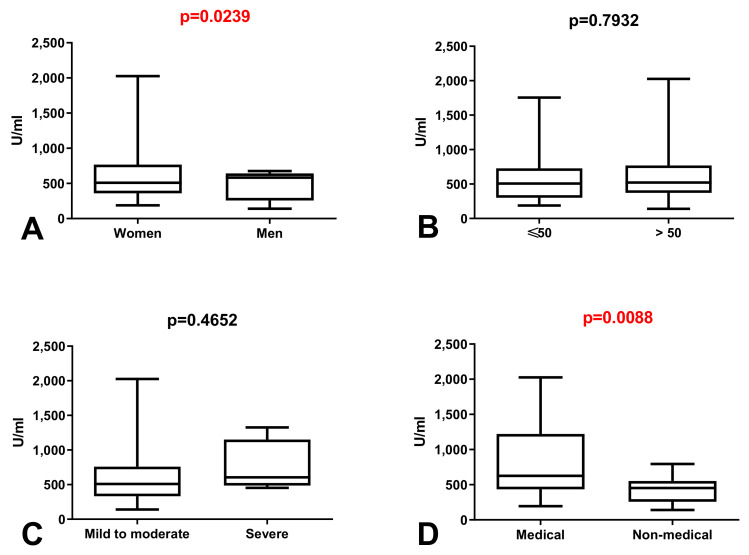
(**A**) Comparison of total anti-SARS-CoV-2 antibodies level between group of women and men without history of COVID-19. The data are presented as median (minimum–maximum). (**B**) Comparison of total anti-SARS-CoV-2 antibodies level between groups of patients before and after the age of 50 without history of COVID-19. The data are presented as median (minimum–maximum). (**C**) Comparison of total anti-SARS-CoV-2 antibodies level between groups of patients with mild to moderate and severe symptoms after vaccination in patients without history of COVID-19. The data are presented as median (minimum–maximum). (**D**) Comparison of total anti-SARS-CoV-2 antibodies level between group of medical and non-medical workers without history of COVID-19. The data are presented as median (minimum–maximum).

**Figure 2 vaccines-09-01508-f002:**
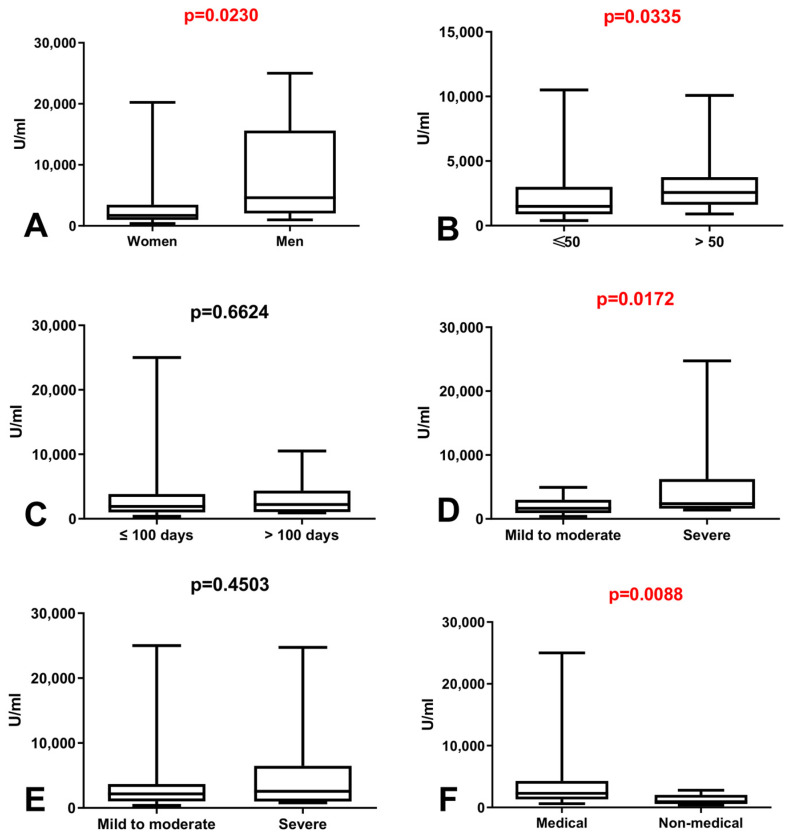
(**A**) Comparison of total anti-SARS-CoV-2 antibodies level between group of women and men with history of COVID-19. The data are presented as median (minimum–maximum). (**B**) Comparison of total anti-SARS-CoV-2 antibodies level between groups of patients before and after the age of 50 with history of COVID-19. The data are presented as median (minimum–maximum). (**C**) Comparison of total anti-SARS-CoV-2 antibodies level depending on the time between positive result of SARS-CoV2 PCR and vaccination date. The data are presented as median (minimum–maximum). (**D**) Comparison of total anti-SARS-CoV-2 antibodies level between groups of patients with mild to moderate and severe symptoms during COVID-19. The data are presented as median (minimum–maximum). (**E**) Comparison of total anti-SARS-CoV-2 antibodies level between groups of patients with mild to moderate and severe symptoms after vaccination in patients with history of COVID-19. The data are presented as median (minimum–maximum). (**F**) Comparison of total anti-SARS-CoV-2 antibodies level between group of medical and non-medical workers with history of COVID-19. The data are presented as median (minimum–maximum).

**Figure 3 vaccines-09-01508-f003:**
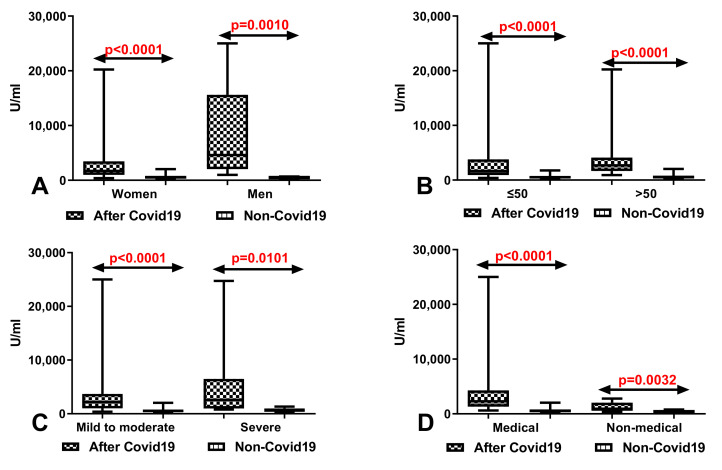
(**A**) Comparison of total anti-SARS-CoV-2 antibodies level in the groups of women and men with and without history of COVID-19. The data are presented as median (minimum–maximum). (**B**) Comparison of total anti-SARS-CoV-2 antibodies level in groups of patients before and after the age of 50 with and without history of COVID-19. The data are presented as median (minimum–maximum). (**C**) Comparison of total anti-SARS-CoV-2 antibodies level in groups of patients with mild to moderate and severe symptoms after vaccination with and without history of COVID-19. The data are presented as median (minimum–maximum). (**D**) Comparison of total anti-SARS-CoV-2 antibodies level in groups of medical and non-medical workers with and without history of COVID-19. The data are presented as median (minimum–maximum).

**Table 1 vaccines-09-01508-t001:** Characteristics of study group.

Parameter	Workers with History of COVID-19 (*n* = 50)		Workers without History of COVID-19 (*n* = 50)	
	Number of Workers (%)	Median (5–95% Percentile) of Anti-SARS-CoV2S Level (Uml))	Number of Workers (%)	Median (5–95% Percentile) of Anti-SARS-CoV2S Level (Uml))
**Age**				
≤50	31 (60%)	1497 (456.9–8293)	24 (48%)	507.5 (189.8–1707)
>50	18 (36%)	2564 (901–10,077)	26 (52%)	522.5 (372.8–770.8)
**Sex**				
Male	9 (18%)	4603 (516.5–10,461)	5 (10%)	583 (257.5–675)
Female	41 (82%)	1705 (398–20,229)	45 (90%)	510 (201.6–1756)
**Type of workers**				
Medical	43 (86%)	2297 (2559–6183)	29 (58%)	626.0 (210–1891)
Non-medical	7 (14%)	941.5 (542.6–1967)	21 (42%)	453 (145.7–791.7)
**Symptoms of COVID-19**				
mild to moderate	32 (64%)	1665 (451–4774)		
severe	18 (36%)	2390 (1397–24,726)		
**Symptoms after vaccination**				
mild to moderate	43 (86%)	2162 (528–18,284)	46 (92%)	510 (190.1–1756)
severe	7 (14%)	2558 (795–24,726)	4 (8%)	604.5 (453–1325)
**Time between PCR** **and 2 dose**				
<100	40 (80%)	1904 (510–24,501)		
>100	10 (20%)	2210 (901–10,504)		

## Data Availability

The full data presented in this study are available on request from the corresponding author.

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
