# Peer review of "Assessment of Post-Vaccination Antibody Response Eight Months after the Administration of BNT1622b2 Vaccine to Healthcare Workers with Particular Emphasis on the Impact of Previous COVID-19 Infection"

_vaccines, 2021, doi:10.3390/vaccines9121508_

Round 1

Reviewer 1 Report

In a manuscript entitled “Assessment of post-vaccination response eight months after the administration of BNT1622b2 vaccine to healthcare workers with particular emphasis on the impact of previous COVID-19 infection” Authors have evaluated post-vaccine antibody titers 8 months after the second vaccine dose had been administered to a group of employees of the Hospital of the Ministry of the Interior and Administration in Olsztyn (Poland). This paper has been prepared with great care and in my opinion, can be published in the present form.

In the time of COVID pandemic the rationale work, like this, should be published asap.

The statistical accuracy and reliability of the research do not raise any objections.  

Author Response

Thank you for revision.

Reviewer 2 Report

The article is reviewed and it is recommended to accept it with minor revision, in the following aspects:

Page 2, line 93, correctly reference the Gene Expert tester, as later if it is done with the Cobas e411 from Roche.

In line 138 on page 4, reference is made to figure 1A, but this does not appear in the job, figure 1 is missing.

On page 6, line 202 speaks of 6 billion people, which is a reference to an American counting method (1 billion is a billion), perhaps it should refer to a European format.

On line 209 page 7 reference is made to the information published  in Szczepienie przeciwko COVID-19 - Szczepienie przeciwko COVID-19 - Portal Gov.pl (www.gov.pl) this is a website of the Polish government and should be explained to what type of document refers

In line 221 page 7, reference is made to a work published in Vaccines, but that explicit reference to the journal is not necessary, it must be cited like the others

In the references of the work there are different and double numbering of the referenced works, the references must be corrected in their numbering.

Author Response

Thank you for your revision. We corrected our article according to your suggestions. All corrections has been marked by red font.

Reviewer 3 Report

Wolszczak-Biedrzycka et al. analyzed antibody levels against SARS-CoV-2 after vaccination with BNT1622b2 vaccine in 100 health care workers with or without prior infection. They demonstrated that those with a history of COVID-19 had significantly higher antibody levels than those without a history. These are consistent with the published study (Zhong et al., JAMA, 2021 doi:10.1001/jama.2021.19996) demonstrating higher antibody levels in 1,960 health care workers after mRNA SARS-CoV-2 vaccination with prior infection than those without prior infection up to 6 months. This manuscript could be improved if the authors address the following points.

  1. Title

Assessment of post-vaccination response” is a bit difficult to understand“; Assessment of post-vaccination antibody response” or other titles including a term of antibody or humoral response could be considered.

  1. Timing of the previous infection

Timing of the previous infection (> 90 days) is known to affect antibody response with prior infection (Zhong et al., JAMA, 2021 doi:10.1001/jama.2021.19996), which is desired to be analyzed in this study if those data were available.

  1. Statistics

”For a normal distribution, the Student's t-test was used. In the case of the lack of normal distribution, the Mann-Whitney U test was used.” is described. Multiple statistical comparisons were performed in the context of different conditions on the population. However, there is no description of how the multiplicity of the comparisons was adjusted. Please clarify the method to adjust multiplicity.

Author Response

(The authors gave the same response as above.)

Round 2

Reviewer 3 Report

The authors addressed all the concerns. The manuscript is acceptable.